# Pharmacokinetic and Pharmacodynamic Drug–Drug Interactions: Research Methods and Applications

**DOI:** 10.3390/metabo13080897

**Published:** 2023-07-29

**Authors:** Lei Sun, Kun Mi, Yixuan Hou, Tianyi Hui, Lan Zhang, Yanfei Tao, Zhenli Liu, Lingli Huang

**Affiliations:** 1National Reference Laboratory of Veterinary Drug Residues, Huazhong Agricultural University, Wuhan 430000, China; sunlei23@webmail.hazu.edu.cn (L.S.); mikun@webmail.hzau.edu.cn (K.M.); hyx97@webmail.hzau.edu.cn (Y.H.); huitianyi@webmail.hzau.edu.cn (T.H.); zl-0118@webmail.hzau.edu.cn (L.Z.); tyf@mail.hzau.edu.cn (Y.T.); 2MAO Key Laboratory for Detection of Veterinary Drug Residues, Huazhong Agricultural University, Wuhan 430000, China; liuzhli009@mail.hzau.edu.cn; 3MOA Laboratory for Risk Assessment of Quality and Safety of Livestock and Poultry Products, Huazhong Agricultural University, Wuhan 430000, China

**Keywords:** drug–drug interaction (DDI), pharmacodynamic interaction, pharmacokinetic interaction, drug combination, cocktail, mathematical model

## Abstract

Because of the high research and development cost of new drugs, the long development process of new drugs, and the high failure rate at later stages, combining past drugs has gradually become a more economical and attractive alternative. However, the ensuing problem of drug–drug interactions (DDIs) urgently need to be solved, and combination has attracted a lot of attention from pharmaceutical researchers. At present, DDI is often evaluated and investigated from two perspectives: pharmacodynamics and pharmacokinetics. However, in some special cases, DDI cannot be accurately evaluated from a single perspective. Therefore, this review describes and compares the current DDI evaluation methods based on two aspects: pharmacokinetic interaction and pharmacodynamic interaction. The methods summarized in this paper mainly include probe drug cocktail methods, liver microsome and hepatocyte models, static models, physiologically based pharmacokinetic models, machine learning models, in vivo comparative efficacy studies, and in vitro static and dynamic tests. This review aims to serve as a useful guide for interested researchers to promote more scientific accuracy and clinical practical use of DDI studies.

## 1. Introduction

Due to the complexity of disease, combination therapy has become a mainstay of treatment. In some cases, however, combination regimens can cause some degree of drug interaction problems due to the large number and variety of drugs the regimens contain. It has been shown that prescription drug regimens contained an average of 6.58 drugs, and these drugs may cause an average of 2.68 drug–drug interactions [1]. The study of drug–drug interactions (DDIs) can help scientifically evaluate pre-selected medication regimens and develop the safest, most effective, and quality-controlled dosing regimens for patients.

DDI is defined as the process by which the pharmacokinetic or pharmacodynamic process of a drug is altered by the influence of another drug after combination. The former is often referred to as “victim” and the latter as “perpetrator”. Pharmacokinetic interactions (PK DDI) often assess the effect of drug–drug interactions by comparing the absorption, distribution, metabolism, and elimination (ADME) processes of the test drug with and without a perpetrator. Pharmacodynamic interactions (PD DDI) are often divided into synergistic, additive, and antagonistic effects, which are judged based on changes in drug effects. The study of DDIs currently involves two main applications: the safety assessment of drug–drug toxicology and the development of compound dosing regimens [2,3]. However, except for some machine learning methods that involve both pharmacokinetic and pharmacodynamic considerations, most basic studies still evaluate DDIs from a single perspective of pharmacokinetics or pharmacodynamics, which leads to some errors in the evaluation results [4,5]. Therefore, the main methods of pharmacokinetic and pharmacodynamic interactions are stated and summarized in this review.

In January 2020, the United States Food and Drug Administration (FDA) issued two guidance standards entitled “Clinical Drug Interaction Studies—Guidance for Industry on Cytochrome P450 Enzyme- and Transporter-Mediated Drug Interactions” and “In vitro Drug Interaction Studies-Cytochrome P450 Enzyme- and Transporter-Mediated Drug Interactions”, which guided studies on drug interactions based on metabolic enzymes and transporters [5,6,7]. In January 2021, the China Food and Drug Administration issued the industry guidance entitled “Technical Guidelines for Drug Interaction Studies (Trial)”, which stipulates that the main contents of DDI study include the following five aspects: whether the drug under study changes the kinetic characteristics of other drugs, whether other drugs can change the kinetic characteristics of the drug under study, assessing the degree of change in the kinetic parameters, assessing the clinical significance of DDI of the drug under study, and the prevention of and control strategy for clinically serious DDI [8]. On the contrary, there are no official regulations and industry guidelines for the assessment of PD DDI so far, and pharmacodynamic studies have not been structured like PK DDI. This summary of pharmacodynamic methods is intended to provide guidance to researchers involved in this area of research.

In this review, the research methods and applications are summarized by focusing on pharmacokinetic and pharmacodynamic interactions. The primary methods for DDI research could be divided into two categories: pharmacokinetic methods and pharmacodynamic methods. PK DDI methods include probe drug cocktail methods, liver microsome and hepatocyte models, static models, physiologically based pharmacokinetic models, and machine learning models. PD DDI methods involve in vivo comparative efficacy studies and in vitro static and dynamic tests (as shown in Figure 1). These serve as reasonable evaluations of combination drugs and are based on changes in efficacy. They reduce the emergence of drug resistance and reasonably and effectively control disease development through DDI research, providing clinicians with rapid and effective compatibility help.

## 2. Pharmacokinetic Interactions

Pharmacokinetic interaction studies focus on the tested drug and the related metabolic enzymes or transporters. The metabolic enzymes or transporters involved in these studies are mainly those from the cytochrome P450 enzyme system, followed by p-gp protein and OATP (Figure 2). These studies mainly focus on evaluating the degree of change and clinical significance of kinetic characteristics with drug combination and determine a reasonable dosing regimen to guide clinical application.

### 2.1. Probe Drug Cocktails 

According to the change in pharmacokinetic parameters of substrates of specific drug metabolic enzymes or transporters, probe cocktails identify the effect of the drug of interest on specific drug metabolic enzymes or transporters. A single cocktail includes a certain number of substrates of different drug-metabolizing enzymes and/or transporters. Thus, it should be validated that there is no interference between various components compared with components being individually administered in order to reduce the impact on the test drug. Current published cocktails mainly involve CYP enzymes, such as CYP1A2, CYP2B6, CYP2C8, CYP2C9, and CYP2C19, and intestinal and liver transporters, such as P-gp and OAT1/3. In addition to these, a few DDI-related studies have been carried out on non-CYP enzymes, namely UDP-glucuronosyltransferases (UGTs). At present, the approach is very valuable for efficiently both verifying in vivo DDI and detecting in vitro DDI, and it can be very useful for bridging in vitro to in vivo data.

#### 2.1.1. In Vitro Cocktail

The FDA guidance recommends that in vivo studies may be exempted when in vitro data show no significant DDI [9]. Various in vitro cocktails have been developed for detecting direct DDI and screening time-dependent inhibition (TDI) [10]. The initial in vitro cocktail was generally developed for direct detection of inhibition of the five major CYP isoforms (CYP1A2, CYP2C9, CYP2C19, CYP2D6, and CYP3A4) [11]. In recent years, CYP2B6 and CYP2C8 have been gradually included according to FDA and EMA guidelines. An in vitro high-throughput cocktail with nine CYP enzymes, which are CYP2A6 and CYP2J2 in addition to the seven conventional CYP isoforms, has been developed. Meanwhile, a new inhibitor of CYP2C8, gemfibrozil 1-O-β-glucuronide, was found to have a strong inhibition of CYP2C8 of >90%, while its selectivity was also found to be high through screening for TDI [12]. Besides CYP-mediated metabolic enzymes, some studies focus on Phase Ⅱ drug metabolic enzymes, such as UGTs [13]. A validated cocktail of the main five UGT enzymes, which include estradiol for UGT1A1, chenodeoxycholic acid for UGT1A3, serotonin for UGT1A6, propofol for UGT1A9/PROG, and zidovudine for UGT2B7/AZTG, has been developed with human and rat microsomes [14]. This cocktail can be utilized to explore UGT inhibition in a high-throughput manner.

The first dedicated in vitro transporter cocktail (digoxin for P-gp, rosuvastatin for BCRP and OATP, metformin for OCT and MATE, and furosemide for OAT) was published to evaluate transporter-mediated DDIs [15]. Furosemide, in addition to being a substrate of OAT1/3, potently inhibits BCRP (50%), enriching background information on furosemide in this study. The preliminary constructed in vitro cocktail also provided data support for the subsequent in vivo cocktail establishment. Simulations using well-established in vitro models and actual in vivo situations sometimes have some differences, and cocktails can simultaneously assess the interaction of multiple transporters and metabolic enzymes in vivo, thereby improving the accuracy of in vitro prediction [16,17]. The extract of goldenseal inhibited the BCRP and OATP1B1/3 in basic models, but it had no effects on the activity of BCRP and OATP1B1/3 with an in vivo cocktail [18]. 

#### 2.1.2. In Vivo Cocktail 

Numerous mature cocktails have been published over the past two decades, such as the Cooperstown cocktail [19]. There are three cocktails specifically used in DDI studies involving CYP enzymes, namely the Inje, Basel, and Geneva cocktails [20,21,22] (As shown in Table 1). Compared with previous developed cocktails, these cocktails reduce the adverse effect profiles and are commercially available. The Inje cocktail can be applied to DDI involving enzyme induction or inhibition, especially gastrointestinal metabolism. The latter two are more recent cocktails and contain more CYP enzymes in comparison with the Inje cocktail, with the Geneva cocktail even including the substrate of transporter p-gp. Recent studies have evaluated the safety of the Basel cocktail in patients with liver injury, which is conducive to timely adjustment of drug use in patients with liver cirrhosis [23]. In addition, the stability of the Basel cocktail was strengthened through deglucuronidation of the plasma samples, thus strengthening its practicability [24]. The safety of the Geneva cocktail has been verified in people from three different geographic origins by Rollason [25]. Meanwhile, this cocktail, combined with a dried blood spots sampling technique, is capable of testing CYP phenotyping and validating DDI with microdosing and low invasiveness.

There is a problem of poor selectivity of substrate in developing transporter cocktails. Metformin is not only the substrate of OCT2 but also the substrate of MATE1/2K [26]. Therefore, it is not clear whether the tested drug has an impact on the single transporter. At present, most studies reduce this disadvantage by optimizing the safety and increasing the number of transporters. Compared to rosuvastatin administered individually, it was found that Cmax and AUC_0-tz_ of rosuvastatin increase by approximately 40% in the previously published cocktail containing digoxin (0.25 mg, P-gp), furosemide (5 mg, OAT1/3), metformin (500 mg, MATE1/2-K), and rosuvastatin (10 mg, OATP1B1/3, BCRP). When the dose of furosemide and metformin is reduced to 1 and 10 mg, respectively, the exposure amount of rosuvastatin is similar to that when administered alone, that is, the mutual DDI is eliminated [27,28]. To avoid harming the health of the subjects, a pre-clinical in vivo model, cynomolgus monkey, was constructed. This model reduces the uncertainty of DDI assessment caused by the differences in K_i_ and the unbound maximum plasma liver inlet concentration [29].

### 2.2. Hepatic Microsomal and Hepatocyte Models 

Hepatic microsomes and cell models were evaluated for drug–drug interactions by assessing activity and mRNA expression levels of metabolic enzymes or transporters. Currently, common liver microsomes include rat, mouse, and human liver microsomes (RLMs, MLMs, and HLMs), and hepatocytes commonly include mouse, rat, monkey, and human hepatocytes [30,31,32]. For application, this model can be utilized to evaluate the DDI caused by TDI and reverse inhibition of enzymes [33]. At the same time, due to the increased attention to drug interactions between drugs and herbs, pharmacokinetic data will also be obtained through this hepatic microsomes and hepatocytes model to preliminarily determine the presence of DDIs [34,35,36]. The advantage of this model is that it is convenient for identifying the metabolic pathways and metabolite species of the tested drugs to verify the DDI related to metabolism-dependent inhibition (MDI) [37]. Debrafenib, a potent ATP-competitive inhibitor of the V600 mutant BRAF kinase, was found to be primarily metabolized by CYP2C8 (56–67%) and CYP3A4 (24%) using liver microsomes and hepatocyte models, and it also was found to induce mRNA expression of CYP3A4 at 30 μM. Desmethyl-dabrafenib was found to weakly inhibit CYP2B6, 2C8, 2C9, 2C19, and 3A4 in metabolic pathway studies [38]. From this, it can be determined that dabrafenib may be an inducer of CYP3A4 and would be a victim of both CYP2C8 and CYP3A4 inhibitors. For transporter-mediated DDIs, OATP1B1 and OATP1B3, which are the important liver-specific transporters, are worth considering. One study compared DDI predictions with physiologically relevant primary hepatocytes and cell lines expressing OATP1B1/3 and found that both yielded similar results. That is, DDI prediction can also be performed on non-physiologically relevant cell lines [39]. Meanwhile, it found that the inhibitory effect could be enhanced when the inhibitor was pre-incubated with physiologically relevant cells.

This method also has certain disadvantages. For example, some drugs have DDI results in vitro but no DDI that occurs in vivo tests, which will lead to the loss of many potentially effective candidate compounds during the screening of new drug leads. At present, some remedies have been developed for this shortcoming. ENG et al. attempted to set a boundary value for the TDI inhibition rate constant obtained from performing TDI assays in hepatic microsomes and hepatocytes to avoid overevaluation of DDIs [40]. Taking the results obtained from hepatocytes and liver microsomes as input parameter values, DDI prediction using static or dynamic models can also reduce the occurrence of overestimation events [41].

### 2.3. Static Model 

The static model is a mechanistic mathematical model that is constructed using minority and readily available parameters. This model simulates the DDI in a steady-state situation [42]. Fold increase in the area under the blood–victim level curve (AUCR) is the preferred parameter of the static model. Iga et al. constructed a simple two-compartment static model for CYP3A4-mediated DDI based on overall inhibitory activity (A_i,overall_) calculated by AUCR [43,44]. In addition, parameters such as oral clearance rate and the proportion of oral drug excretion through liquid were introduced to improve the accuracy of model prediction. The static model can be used to analyze the characteristics of DDI under fixed conditions and provide parameter selection for the subsequent construction of dynamic models, as well as local screening of drug compatibility schemes [45].

The static model can also be applied to the study of DDIs involving multiple metabolic enzymes or transporters, that is, adding the data of each metabolic enzyme or transporter involved to evaluate DDI at a certain time and state. It can be used to analyze the mechanism of drug interaction. At present, the static model can be used to evaluate the effect of individual gene polymorphism on drug interaction. Individual gene polymorphism mainly refers to a small number of individuals with different phenotypes of CYP enzymes or transporters that result from individual gene changes. Compared with the rest of the population, this small group of individuals can experience serious drug interactions following test drug ingestion. The most dangerous interactions occur as a result of slow metabolism, drug accumulation, and increased toxicity. The structure and principle of the static model are simple, so it is easy to explore the mechanism [46,47].

The static model also has an inherent disadvantage. It only considers the drug interaction at the steady-state concentration, thus leading to overestimation of risk. Selecting this model for drug interaction risk assessment requires increasing the correction factor or adding some mechanistic parameters to achieve more accurate prediction results. Taguchi et al. added kinetic parameters of deactivation and recovery of OATP transporter to the static models R2 and R3, which reduced the error of in vitro simulation and improved the authenticity and accuracy of the simulation [48].

The static model also has disadvantages when compared with the dynamic model. The dynamic model can estimate the pharmacokinetic changes at any time and obtain results that are more in line with real body situations [49]. Peters et al. constructed dynamic models using Simcyp, considering the rate and degree of metabolite penetration in the intestine, so that the results obtained reflect truer values than static model results [50]. After comparative experiments between dynamic and static models, it was found that a better overall assessment can be obtained and the accuracy of prediction can be greatly improved when the static model is applied in combination with the dynamic model for the assessment of drug interactions [51]. 

### 2.4. Physiologically Based Pharmacokinetic Model 

Physiologically based pharmacokinetic models (PBPKs) have been gradually expanding in scope of application since their introduction by Theroll in 1937 [52]. Because static models are often overestimated and PBPK models have physiologically relevant and more realistic pharmacokinetic simulations, the latter are more suitable for accurate DDI analysis [53]. Currently, drugs that only use the PBPK model for DDI prediction without in vivo testing are mainly victims of DDI, and only a few perpetrators can only use this method for DDI prediction. It can be found from studies on metabolic enzymes and transporters in recent years that CYP3A4/5, P-gp, and OATPs are of greater concern [54,55]. The commonly used software for constructing PBPK models to predict DDI are GastroPlus and Simcyp simulators.

In the construction of a DDI-PBPK model, the PBPK model of the test drug is usually built first, and then the preliminary model of the drug is optimized by experimental data. The PBPK model of the drug interaction is constructed in the same way. The two PBPK models are combined by relevant parameters to evaluate DDI (Figure 3). In PBPK modeling, the enzymes or transporters that may be affected by the drug to be tested through literature or clinical trials should be first discerned, so as to carry out targeted simulation evaluation [56].

#### 2.4.1. DDI-PBPK Associated with Metabolic Enzymes 

As mentioned earlier, CYP enzymes are the focus of daily assessment of DDIs associated with metabolizing enzymes, with considerable concern for CYP3A4/5, CYP2C8, and CYP2B6 [58,59]. Esaxerenone, a selective mineralocorticoid receptor blocker, was demonstrated to be a time-dependent inhibitor and inducer of CYP3A by in vitro assays. Time-dependent changes in enzyme activity could be predicted accurately, and esaxerenone was also predicted to have a low level of DDI potential by a PBPK model [60]. The impact of protein polymorphisms on DDI prediction is assessed analytically by adding information from proteomics. A population-based PBPK model was developed to predict the DDI between efavirenz and lumefantrine and included the polymorphic information of CYP2B6. The results showed that compared with *1/*1 population (wild type), DDI caused by the two drugs was aggravated in people expressing *6/*6, that is, 2–3-fold higher efavirenz plasma concentration [61]. 

Specific pharmacokinetic parameters of inhibitors (or inducers) and physiological parameters related to the metabolic enzymes are capable of improving the accuracy of PBPK models. Yamada et al. evaluated 17 drugs with minor harmful effects on intestinal CYP3A substrates and found that the occurrence of false negative results was reduced and the accuracy of the PBPK model for DDI prediction was improved by inclusion of the parameters related to TDI, reversible inhibition, induction and mechanism, and selection of the concentration–time profile in enterocytes [62]. The hepatic accumulation factor of an inhibitor (Kp_uu,liver_) was selected as the input parameter of a PBPK model, which promoted the prediction ability of the model. 

Itraconazole, as a typical inhibitor of CYP3A4, could be directly utilized as a tool in predicting the effects of a drug associated with CYP3A4, if a PBPK model for itraconazole exists, without further model development and fitting [63,64]. Chen et al. constructed a PBPK model of itraconazole and its major metabolite, OH-ITZ. The model can accurately predict the drug interactions related to CYP3A4 [65]. In future research in this area, PBPK modeling will be broadly adopted. If we can establish some representative PBPK models of substrates, inhibitors, or inducers to form a model library in advance, it would be of great help to future research. 

#### 2.4.2. Transporter-Associated DDI-PBPK 

The involvement of drug transporters is often required during drug absorption, distribution, and elimination. The current PBPK model primarily assesses DDIs mediated by two transporters, namely P-gp and OATP1B1/3 [66,67]. The PBPK model of drug interactions associated with metabolic enzymes is modeled similarly, i.e., an optimized PBPK model using interaction data or a combination of two established PBPK models for assessment of drug interactions.

It was possible to judge whether the tested drug would have an effect on other drugs (eliminated and distributed by the same transporter) based on transporter abundance changes. Yamazaki et al. not only used the data of clinical combination to optimize the model but also added the parameters of P-gp abundance to judge whether there would be serious DDI [68]. Similarly, selecting an endogenous marker as the subject of a PBPK model, the amount of which will be greatly increased or decreased after drug interaction, indirectly assesses the risk of DDI [69].

It was found that CYP isoforms and P-gp (which can transport intracellular drugs to the extracellular space) or CYP isoforms and OATPs (hepatic uptake pathways of drugs) are mainly included. Asaumi et al. constructed a PBPK model of rifampicin to assess the risk of DDIs involving saturated uptake by the liver and active induction of metabolic enzymes [70]. Enriching the modeling data and integrating the drug data obtained in vivo, in vitro, and clinically ensured the accuracy and authenticity of the model. An integrated in silico, in vitro, and clinical approach—including an inhalation PBPK model—successfully avoided any potential DDI risks with nemiralisib [71]. Chen et al. assessed the effect on drug interactions involving CYP and P-gp substrates when patients underwent gastric bypass surgery [72].

In a pharmacokinetic interaction study, the cocktail method can quickly screen out many metabolic enzymes or transporters that may be involved in the DDI between the test drugs, saving cost and time. When the mechanistic mathematical model is used for DDI evaluation, the number of subjects or test animals can be reduced, lowering its error range and increasing its accuracy, because the mathematical model needs to be fitted and modified many times. The key question is whether the selection of parameters is reasonable and consistent with the drug action process in the actual animal body. The liver microsome model can directly obtain the data of DDI in vitro, but there are still some defects regarding whether the metabolic process in vitro is consistent with the actual process in vivo with minimized error.

### 2.5. Machine Learning Model

In view of the disadvantages of high consumption time in clinical trials and poor correlation of in vitro tests, the best alternatives are computation methods, which can be used to judge whether DDI occurs in combination with drugs by learning the DDI and drug information that is now available. In recent years, numerous machine learning models have been developed for assessment prediction of DDIs. The knowledge graph (KG) embedding model with DDI and its side effects as learning tasks is one of them. By introducing adversarial autoencoders, the model generated more reasonable negative datasets, thereby improving the predictive ability of the model; by applying Gumbel-Softmax relaxation and Wasserstein distance, the problem of gradient disappearance on discrete data is solved and the convergence rate of the KG embedding model is accelerated [73].

Drug label information included in DrugBank and available through the FDA was integrated as training and validation datasets for machine learning. The PK DDI prediction model was based on the changes in pharmacokinetic parameters as the learning task and regression bagged tree was used to establish the model. Based on this, an independent application model containing drug information, single-nucleotide polymorphism, and drug recommendation information based on anatomical therapeutic chemical level is proposed. In this model, drug pairs from DrugBank were presented by polypeptide-PD-Drug-Type, where each DDI pair was encoded as a 2830-dimensional vector, and thereby, a tree containing 615 branches was built [74]. A similar approach is few-shot learning, which is one that requires learning new techniques and capabilities from only a small amount of label data [75].

A drug is a single entity composed of multiple functional groups, so the subsequent PK and PD are determined by the substructures contained in each. Therefore, DDI studies of new drugs without clear label information can also be accurately predicted [76]. According to the interaction score of substructure pairs obtained by the substructure co-attention model, the accuracy of DDI prediction can be improved. The predicted ACC, AUC, AP, and F1 of the model, respectively, reached 68.57 ± 0.30, 74.96 ± 0.40, 75.44 ± 0.50, and 65.32 ± 0.23 between new drugs and new drugs. The predicted ACC, AUC, AP, and F1 of the model, respectively, reached 77.72 ± 0.30, 84.84 ± 0.15, 84.87 ± 0.40, and 78.29 ± 0.16 between new drugs and old drugs [77].

In addition to the above network-based computer methods, there are also quantitative structure–activity relationships (QSARs)-based methods. By investigating the interaction between drug molecules and targets of action, it was found that different molecular structures and sizes produce specific DDIs for different targets of action [78]. Molecular weight (MW) is a screening condition to distinguish substrates of CYP3A4 (MW ≥ 360 g/mol) and other CYP enzymes [79]. Utilizing a Support Vector Machine based on fingerprints and d eep learning models showed positive predictive values >80% based on the endpoint of MW.

## 3. Pharmacodynamic Interactions

Pharmacodynamic interactions include three types: synergistic, additive, and antagonistic. Synergism and additivity are effective in controlling disease development without causing serious side effects, whereas antagonism is the opposite [67,80]. Studies of pharmacodynamic interactions are often focused on central nervous system drugs, antiparasitic drugs, and antibiotic drugs. The study of DDI makes it fast and simple to formulate a rational dosing regimen from the perspective of pharmacodynamic interaction, but this is not so with the study of mechanisms. Therefore, the determined dosage regimen according to pharmacodynamics has certain limitations; for instance, different regions or populations will produce different results for a certain dosage regimen.

The main idea of the method is to compare the changes in efficacy between combination and single-drug regimens or two different compatibility regimens, so as to determine whether one of the regimens plays a positive role in controlling the development of the disease and ensuring life and health. At present, two common methods are in vivo comparative efficacy studies and in vitro static and dynamic models.

### 3.1. In Vivo Comparative Efficacy Studies

An in vivo comparative pharmacodynamic study assesses drug interactions by comparing physiological indicators after combination therapy and monotherapy. This method is mainly used to observe whether there is a mutual influence between drugs macroscopically and is commonly used for the evaluation of anesthetic and analgesic drugs.

The combination of narcotics and analgesics is commonly assessed using an in vivo approach. The feasibility of this combination is determined by comparison with results from a positive control group [81,82]. Stemmet et al. compared a combination of anesthetics, analgesics, and sedatives with preferred etorphine and azaperone, using invasive blood pressure and blood gas analysis [83].

When two analgesics are used in combination, the value of that combination is based on pain assessment methods and measurement of the nociceptive pain threshold. Bustamante et al. chose the Glasgow composite pain scale (*p* < 0.01) and pain rating scale to evaluate analgesic effect [84]. In addition to anesthetics and analgesics, the comparative study method is suitable for other drugs [85,86]. Sajedeh et al. assessed four non-steroidal anti-inflammatory drugs (NSAIDs) and steroidal anti-inflammatory drugs with regard to pathological changes and inflammatory factor levels [87].

In vivo comparative studies can be used to evaluate and compare drug combinations based on macroscopic changes such as clinical response and physiological and biochemical parameters of subjects, but the selection of evaluation criteria may affect the evaluation results. It is often necessary to have known effective combination drug regimens for comparison, though the number of known drug regimens is small, which further limits the use of this method. However, as a method for primary screening of combination drugs, it is still of value.

The drug dose ratio may also affect the results of drug combination, that is, different proportions of components in the dosing regimen will produce different pharmacodynamic responses, such as synergy and antagonism [88,89]. Current clinical regimens for combination drugs include fixed-dose combination medicinal products (FDCs), such as analgesic and antihypertensive drugs [90,91]. Such combination regimens are based primarily on the dosing ratio determined by the maximum tolerated dose, with a wider range of active effects and higher efficacy, as well as positive feedback of patient populations who are insensitive to the individual components in the component [92]. For non-fixed-dose administration regimens, checkerboard methods, mathematical models, and other methods are also used to determine the appropriate administration regimen. Jadhav et al. used dose–response model fitting to predict the dose ratio of each component when bempedoic acid and different statins were combined. Compared with the reduction of low-density lipoprotein cholesterol by 80 mg conventional atorvastatin or 40 mg simvastatin, the combination of 20 mg atorvastatin or 10 mg simvastatin with 180 mg benzoic acid could achieve fourfold results [93].

### 3.2. In Vitro Static and Dynamic Testing

In vitro static and dynamic tests consist in direct observation of the inhibitory effect of two drugs on pathogens in vitro. The drug concentration in static testing is fixed to observe the effect of drugs on pathogens at different set concentrations, while dynamic testing utilizes specific instruments and equipment, such as a hollow fiber model, to monitor the effect of drugs on pathogens under dynamic drug concentrations. These methods are mostly used for combinations of antiviral drugs, antibiotic drugs, or antineoplastic drugs.

Checkerboard methods and bactericidal curves are often chosen for antibiotic static testing to determine whether a combination produces synergism or antagonism [94,95,96]. The change in minimum inhibitory concentration (MIC) is key in assessing the effect of antibiotics. The checkerboard method can determine MIC with only a trace amount of drugs and bacterial solution, making it easy to obtain the MIC of combined drugs and determine the change in drug efficacy [97,98].

The DDI analysis of the combined application of antiviral drugs or antitumor drugs should not only consider the killing ability of target cells but also ensure the growth and reproduction of normal cells. The effects of combined drugs on MSC and spos-2 cells were considered [99]. At the same time, the cell test can also guide setting the administration scheme. The drug concentration in the compatibility scheme is adjusted according to the cell test, which inhibits virus replication on the premise of ensuring the safety of the body [100].

The results of static tests can confirm whether the efficacy of drugs against pathogens is enhanced or weakened at a certain concentration, but mimicking drug processing in the body is still a challenge. Results obtained by relying only on inhibition at several points are questionable. The use of a hollow fiber model can depict the dynamic changes in drugs and pathogens in the body (Figure 4). Broussou et al. cultured Staphylococcus aureus (*S. aureus*) using a hollow fiber model to produce biofilms, which were used to investigate the efficacy of amikacin alone and in combination with vancomycin. In dynamic tests, the combination of the two had little synergy against *S. aureus* biofilm forming when compared with amikacin alone but had a strong synergy for selecting resistance and proliferation of suspended bacteria, thus also reducing resistance to amikacin [101]. Broussou et al. also investigated the effect of a static bactericidal curve and a dynamic bactericidal curve on the results. In the static bactericidal test, the antibacterial effect of vancomycin and amikacin on *S. aureus* was close to the same when used in combination and alone [102]. However, in the dynamic test, vancomycin alone had no bactericidal effect, but the combination significantly reduced the proliferation and expansion of bacteria.

### 3.3. Other Methods

When two anthelmintics are used in combination, larval culture tests are often required to assess their plausibility [103]. Kotze et al. incubated *Haemonchus contortus* with abamectin or monepantel alone or in combination through larval culture tests and found that they were resistant to abamectin and monepantel, respectively, when two groups of worms were incubated alone, and the number of abamectin-resistant parasites decreased when the dose ratio of *abamectin:monepantel* gradually decreased after combined culture. Synergism was found at a specific ratio based on parasite incubation results [104].

The combination of two or more drugs from the perspective of pharmacodynamics was considered as described above. In some specific cases, it is necessary to use a single method for evaluation and comprehensive analysis. There are some differences between the medication regimen and the specific process in vivo from the pharmacodynamic perspective. It is also necessary to better understand the mechanism of successful in vivo testing to flexibly carry out group deduction and maximize the benefits of a compatibility regimen.

## 4. Prospects

Drug interactions are common in combination therapy. The mechanistic study of DDIs can avoid medication risks in patients, reduce the incidence of drug resistance, and maximize the drug effect. Drug–drug interaction research on a single level has its inherent defects; therefore, pharmacokinetic and pharmacodynamic synchronization research will become the predominant trend in accurate DDI prediction.

It is worth noting that the current methods gradually tend toward computer mathematical models, but due to the lack of experimental data, the current model prediction ability is limited. Therefore, it is still necessary to carry out a large number of in vitro or in vivo tests to enrich the relevant databases in order to improve the model prediction ability. For PD DDIs, official regulations and industry guidelines are still the key to future DDI studies. Standardized PD DDIs make DDI studies more accurate and comprehensive. The future methods for DDI need to be improved by focusing on the organic combination of in vitro tests, in vivo tests, and AI to optimize combination drug treatment for patients.

## 5. Study Highlights

### 5.1. What Is the Current Knowledge on the Topic?

In the context of drug combinations, studying drug interactions has become a mandatory research process in the development of new drugs and multidrug combinations. Current studies mostly focus on drug interaction studies in a single aspect, that is, pharmacokinetic interactions or pharmacodynamic interactions, and few studies with both have emerged.

### 5.2. What Question Did This Study Address?

This study collected study methods related to drug interactions to assist in subsequent related studies.

### 5.3. What Does This Study Add to Our Knowledge?

In this paper, five pharmacokinetic methods and three pharmacodynamic methods are summarized, and the advantages and disadvantages of various research methods are summarized.

### 5.4. How Might This Change Clinical Pharmacology or Translational Science?

By summarizing the pharmacokinetic and pharmacodynamic drug interaction research methods, a more complete research program can be proposed, and then, a more scientific and reasonable theoretical basis can be proposed for the combination of drugs to ensure the safety of the medicated population.

## Figures and Tables

**Figure 1 metabolites-13-00897-f001:**
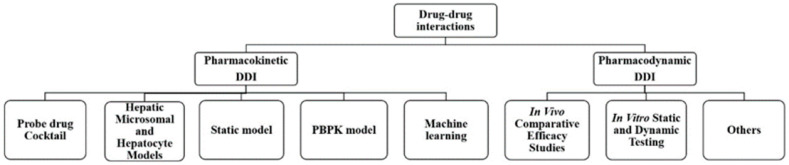
The main methods for drug–drug interactions research.

**Figure 2 metabolites-13-00897-f002:**
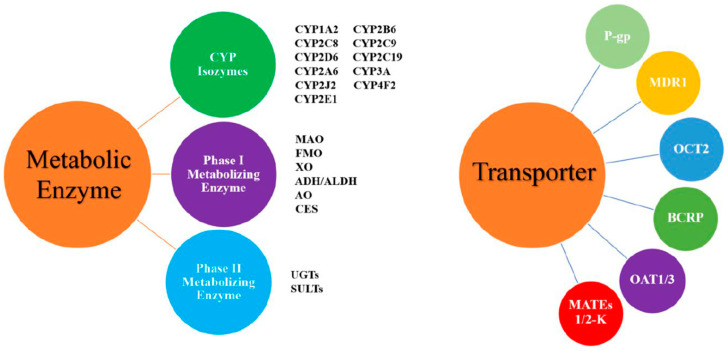
The common drug-metabolizing enzymes and transporters. Abbreviations: ADH/ALDH: alcohol/aldehyde dehydrogenase; AO: aldehyde oxidase; BCRP: breast cancer resistance protein; CES: carboxylesterase; CYP: cytochrome P450 enzyme; FMO: flavin monooxygenase; MAO: monoamine oxidase; MATEs: multidrug and toxic compound efflux transporters; MDR1: multidrug resistance protein 1; OATP: organic anion transporting polypeptide; OAT: organic anion transporter; OCT: organic cation transporter; P-gp: P-glycoprotein; SULTs: sulfotransferases; UGTs: uridine diphosphate glucuronosyltransferases; XO: xanthine oxidase.

**Figure 3 metabolites-13-00897-f003:**
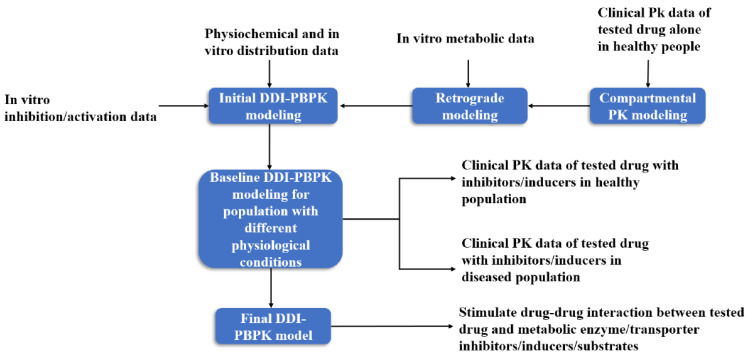
DDI-PBPK model construction workflow [57].

**Figure 4 metabolites-13-00897-f004:**
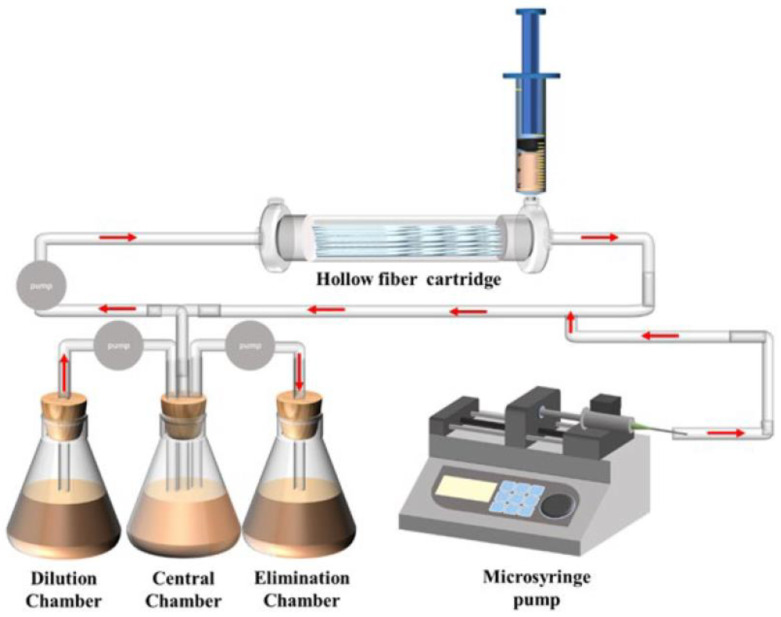
Diagrammatic representation of the hollow fiber infection model courtesy of FiberCell Systems.

**Table 1 metabolites-13-00897-t001:** Three common cocktails used for in vivo DDI prediction.

Cocktail	Probe Drug	Enzyme	Dosage (mg)
Geneva Cocktail	Caffeine	CYP1A2	50
	Bupropion	CYP2B6	20
	Flurbiprofen	CYP2C9	10
	Omeprazole	CYP2C19	10
	Dextromethorphan	CYP2D6	10
	Midazolam	CYP3A4	1
	Fexofenadine	P-glycoprotein	25
Basel Cocktail	Caffeine	CYP1A2	10
	Efavirenz	CYP2B6	50
	Flurbiprofen	CYP2C9	12.5
	Omeprazole	CYP2C19	10
	Metoprolol	CYP2D6	12.5
	Midazolam	CYP3A	2
Inje Cocktail	Caffeine	CYP1A2	93
	Losartan	CYP2C9	30
	Omeprazole	CYP2C19	20
	Dextromethorphan	CYP2D6	30
	Midazolam	CYP3A	2

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
