# Peer review of "Pharmacokinetic and Pharmacodynamic Drug–Drug Interactions: Research Methods and Applications"

_metabolites, 2023, doi:10.3390/metabo13080897_

Round 1

Reviewer 1 Report

The author should normalize the manuscript, for example, the table in Figure 1 is inconsistent with the description in the article, line 318 does not have italics in vitro, etc.

The English expression in the manuscript is not exact, and the author should seek native English speakers to polish it.

Author Response

Dear reviewer,

Thank you for your comments on this review. All authors of this article have appropriately modified the content of the article based on your proposed revisions. The revised content has been uploaded as an attachment for your reference. Please see the attachment.
Thank you for your advice. We look forward to your reply.

 Your sincerely,

Lingli Huang

Reviewer 2 Report

With pleasure, I read the paper titled: “Pharmacokinetic and Pharmacodynamic Drug-Drug Interactions: Research Methods and Applications”. Overall, the paper reads very well. The topic is clinically relevant to the reader of Metabolites journal. Collectively, the article is well-written with good flow of ideas, proper English language, up-to-date citations, and focused tables and figures. The authors adequately introduced the topic and provided rationale for the study. The results are written in simple way and data are summarized in pertinent tables and figures. I have the following comments:

1. Title. I recommend adding the following at the end for completion “a narrative review” to highlight the type of study.

2. Introduction. Please clearly indicate the significance of the study by highlighting how is it different from the previously published literature and how it fills the literature gap.

3. Discussion. Please highlight the limitations of the present study.

Author Response

Dear reviewer,
Thank you for your comments on this review. All authors of this article have appropriately modified the content of the article based on your proposed revisions. The revised content has been uploaded as an attachment for your reference. Please see the attachment.
Thank you for your advice. We look forward to hearing from you.

Your sincerely,

Lingli Huang

Reviewer 3 Report

The paper is OK. It's publishable, with minor adjustments. 

I have noticed that throughout the entire document, in all the models under discussion, whether they are static or dynamic, the pharmacokinetic component predominates.

It is true that pharmacodynamic aspects are more difficult to highlight, especially in vivo, but their interactions can be dangerous or fatal. It is important to note that a drug, even if it is unique, once introduced into the body, already interacts with a myriad of compounds (Tallarida, 2000).

Proportion is very important because two medicinal substances, in different proportions within a combination, can have different efficacies (synergism, antagonism), especially when one of them may exhibit the phenomenon of hormesis. In such situations, we can no longer speak of the same predictions.

Pharmacodynamic interactions, as a suggestion, would be worth mentioning, as well as the quantitative analysis of fixed drug combinations, since an industrial product can contain a single active substance or a binary or multiple combination, but the combinations are always in a fixed proportion.

Please add a small section dedicated to the pharmacodynamic interactions.

Author Response

(The authors gave the same response as above.)

Reviewer 4 Report

The authors revise the mechanisms and methods for the study of drug-drug interactions, both of pharmacokinetic or pharmacodynamic origin.

Several methods are commented, which can be useful to consider when neede to study this conditions mainly for a new drug development.

The abstract is not clearly representative of what is the paper dedicated to. It should be clearer and more precise, substituting the vague introduction paragraphs for more accurate and efficient text ro describe the aim of the revision, what can be found and what for.

The number of references is quite extense and many of them really old. Please revise the need to include them.

minor spelling or edting is requires in some section.

Author Response

Dear reviewer,
Thank you for your comments on this review. All authors of this article have appropriately modified the content of the article based on your proposed revisions. The revised content has been uploaded as an attachment for your reference.
Thank you for your advice. We look forward to your reply.

Your sincerely,

Lingli Huang
